# Trends and Urban-Rural Disparities of Energy Intake and Macronutrient Composition among Chinese Children: Findings from the China Health and Nutrition Survey (1991 to 2015)

**DOI:** 10.3390/nu13061933

**Published:** 2021-06-04

**Authors:** Jian Zhao, Lijun Zuo, Jian Sun, Chang Su, Huijun Wang

**Affiliations:** 1Department of Epidemiology and Health Statistics, Institute of Basic Medical Sciences, Chinese Academy of Medical Sciences, Beijing 100005, China; zhaojian131023@163.com; 2Aerospace Information Research Institute, Chinese Academy of Sciences, Beijing 100101, China; zuolj@radi.ac.cn; 3School of Public Health and Management, Ningxia Medical University, Yinchuan 750004, China; sjnxykdx0202@126.com; 4National Institute for Nutrition and Health, Chinese Center for Disease Control and Prevention, Beijing 100050, China; wanghj@ninh.chinacdc.cn

**Keywords:** children, urban–rural disparity, nutrition transition, macronutrient, dietary reference intake

## Abstract

The nutrition status of children is gaining more attention with a rapid nutrition transition. This study aimed to investigate trends and urban-rural differences in dietary energy and macronutrient composition among Chinese children. A total of 7565 participants aged 6 to 17 years were obtained from three rounds (1991, 2004 and 2015) of the Chinese Health and Nutrition Survey (CHNS). The individual diet was evaluated via three consecutive 24-hour dietary recalls and compared with the Chinese Dietary Reference Intakes (DRIs). From 1991 to 2015, there was a significant increase in children’s fat intake, the proportion of energy intake from fat, and the proportion of children with more than 30% of energy from fat and less than 50% of energy from carbohydrates (*p* < 0.001). Compared with the DRI, the proportion with higher fat and lower carbohydrate intakes were, respectively, 64.7% and 46.8% in 2015. The urban-rural disparities in fat and carbohydrate intake gradually narrowed, while the gap in protein intake increased notably over time (*p* < 0.001). Chinese children experienced a rapid transformation to a low-carbohydrate and high-fat diet. Urban-rural disparities persistently existed; further nutritional interventions and education were of great significance, so as to ensure a more balanced diet for Chinese children.

## 1. Introduction

Nutritional status during childhood is crucial for health, wellness and cognitive development, as well as the future of the whole population. Unhealthy childhood nutrition increases the risk of future non-communicable diseases (NCDs), such as cardiovascular diseases, type 2 diabetes and cancers, and result in high morbidity and mortality worldwide [1]. Nutrient intake is an important implication for the nutritional status of children. The Chinese Dietary Reference Intake standard (DRI) has been used as a standard to determine the recommended dietary intake levels for Chinese people [2]. According to the DRI (2013) for Chinese children, the adequate intake of fat, protein and carbohydrates should account for 20 to 30%, 15% and 55 to 65% of total energy, respectively [3]. Appropriate macronutrient intake such as dietary fat intake in children has been shown to reduce the risk of NCDs, while excessive fat intake in children is associated with the risk of cardiovascular disease in adults [4]. Meanwhile, insufficient energy and protein is one of the most proximal and immediate determinants of child protein-energy malnutrition, which affects not only physical growth but also impaired immune system function, susceptibility to illness and cognitive deficits. Moreover, paradoxically, co-existing with under-nutrition, a global epidemic of overweight and obesity in children is taking over many parts of the world, and China is no exception. With the rapid growth of China’s economy, overweight and obesity among children have increased rapidly in the past four decades, and the latest national prevalence estimates for 2015 to 2019, based on Chinese criteria, were 11.1% for overweight and 7.9% for obesity in children aged 6 to 17 years [5,6]. As the biggest developing country in the world, children in China make up 14.8% of the population. Therefore, nutrition and health of Chinese children form an indispensable and important aspect of national sustainable development [7].

Before the economic reform, a large urban-rural gap existed in China because of the existence of many policy tendencies that favored urban residents. Since 2002, the Chinese government has vigorously promoted rural economic development and construction to make people of different backgrounds have more equitable access to various foods [8]. Previous studies have documented the trends and gaps in health disparities of Chinese rural versus urban adults [9]. Nutrition was also found to be one of the most important factors contributing to health disparities between urban and rural areas. Since children were in the early and critical stage of their physical development, the impact of the nutrition gap between urban and rural areas on children was expected to be more serious [10]. Hence, it is important to provide evidence on the determinants of child health outcomes separately for urban and rural areas, so that fundamentally different nutrition education and interventions can be designed to optimally improve the nutritional status of children in their respective settings [11].

Previous studies have assessed trends in energy and macronutrient intakes and distribution among adults [12], the elderly [13] or young children aged two to six years in China [14]. These studies demonstrated that, despite significant improvements in overall nutritional status and dietary quality, Chinese adults, young children and the elderly are experiencing a rapid nutritional transition to a high-fat, low-carbohydrate diet. Researches on the urban-rural disparities in health and nutrition status have been conducted in central areas [15] or mainly focus on adults in China [8,16]. Meanwhile, a previous longitudinal study has investigated degrees of nutritional disparities between urban and rural children in China during the period 1989 to 2006 using height for-age z-score, weight-for-age z-score, and anthropometric measurements [10]. However, to our knowledge, limited studies have compared secular trends and urban-rural differences in energy intake and macronutrient composition in Chinese children aged 6 to 17 years, and how these disparities have changed over time.

In consideration of the above knowledge gaps, the main aims of this study are to: (i) examine secular trends in dietary energy and macronutrients intakes and macronutrient distribution in the diet of children in China during 1991 to 2015; (ii) explore secular trends on the urban-rural disparities of children’s nutrition and provide evidence and suggestions for child nutrition improvement in China.

## 2. Methods

### 2.1. Design and Participants

Using data from the CHNS, an ongoing prospective study aimed to investigate the impact of the social and economic transformation of Chinese society on the health and nutrition status of its population [17]. The CHNS survey has been conducted every two or three years. A multistage, random-cluster approach was used to draw households from urban and rural areas. Additional details regarding the CHNS are provided elsewhere [18]. In order to observe and compare the degree of transformation and trends at intervals of more than a decade, the present analysis was based on three rounds (1991, 2004 and 2015) of CHNS survey data. After excluding participants who were missing key variables (*n* = 76) and those with extreme values (*n* = 64), 7565 participants aged 6 to 17 years were recruited in a survey year.

### 2.2. Dietary Assessment

Individual diet was assessed via three consecutive 24-h dietary recalls (two weekdays and one weekend day) in each wave of the CHNS and were collected by trained investigators who weighed all the available foods in the participants’ homes [19,20]. The mean average energy intake (kcal/day), fat intake (g/day), carbohydrate intake (g/day), and protein intake (g/day) were derived from dietary survey data from CHNS and calculated based on the Chinese food composition tables (FCTS) [21].

### 2.3. Socio-Demographic Data Anthropometrics

For the present study, the age of children was divided into two stages (6 to 11 years and 12 to 17 years) according to the age period of entering primary school and middle/high school in China. The geographical region was assigned as rural or urban, covering all the corresponding provinces in the CHNS survey. Participants’ height (SecA 260) and weight (SecA 880) were measured by trained health workers and BMI was further calculated (weight (kg)/square of height (m^2^)) [22]. Participants reported all physical activities (PA), and we converted the time spent in each activity into a metabolic equivalent of task (MET) hours per week based on the Compendium of Physical Activities [23]. As for the households’ characteristic, the highest parent education attainment was classified as primary school (or illiterate), middle school, or high school (or above). By adjusting the consumer price index, the per capita annual income in each survey was inflated to the value in 2015. Smoking and drinking status were classified into nonsmokers/smokers and nondrinkers/drinkers, respectively.

### 2.4. Statistical Analyses

SAS 9.4 statistical software (SAS Institute, Cary, NC, USA) was used for all statistical analyses. Continuous variables were represented by adjusted mean and standard errors (SE) and categorical variables by percentage and SE of percentage. The chi-square test was used to assess the association between the different levels (below, meeting and above the recommendations) of DRI for macronutrient. Mixed effect models were developed to calculate the adjusted intake of dietary energy, macronutrient and macronutrient-energy percentages, and to estimate the temporal trends after adjusting for intra-class correlation within clusters and covariates. The energy and macronutrient composition were treated as continuous variables and as the dependent variable, respectively, in each model. A two-tailed *p* < 0.05 was considered statistically significant.

## 3. Results

### 3.1. Social-Demographic Profile

The social-demographic profile of participants in three rounds of CHNS (1991, 2004, 2015) is shown in Table 1. The sample size was 2887 in 1991, 1979 in 2004 and 2699 in 2015, respectively. From 1991 to 2015, the proportion of children aged 6 to 11 years increased from 51.2% to 57.4%, the proportion of girls increased from 49.3% to 52.3%, and the proportion of urban participants increased from 26.3% to 34.5%. Meanwhile, the average BMI of children increased from 17.1 kg/m^2^ to 18.5 kg/m^2^, while the PA decreased from 85.0 MET/s to 70.2 MET/s. In addition, healthier lifestyle choice, greater than primary school education and higher household income became more prevalent over time in the household members (*p* < 0.001).

### 3.2. Secular Trends in Energy Intake and Macronutrient Composition

As shown in Table 2 and Table 3, the dietary energy, protein and carbohydrate intakes of Chinese children showed a downward trend in all age groups, gender and geographical regions (*p* < 0.001). Specifically, the average dietary energy, protein and carbohydrate intake decreased from 2205.3 kcal/day, 63.5 g/day and 369.8 g/day in 1991 to 1671.6 kcal/day, 51.1 g/day and 247.6 g/day in 2015, respectively. However, the average dietary fat intake increased notably from 55.6 g/day in 1991 to 66.2 g/day in 2015. In 2015, urban boys aged 12 to 17 years had the most dietary fat intake (80.1 g/day), and rural boys aged 6 to 11 years had the lowest intake of dietary protein (43.4 g/day) and carbohydrates (185.6 g/day). From 1991 to 2015, the % of energy from fat in Chinese children aged 6 to 17 years increased from 24.2% to 36.8%, whereas the % of energy from carbohydrates decreased from 63.4% to 51.4%. In 2015, urban children aged 6 to 11 years had the most % of energy from fat (36.7% in boys and 37.8 in girls) but the least % of energy (49.9% in boys and 48.9% in girls) from carbohydrates.

### 3.3. Trends of Urban–Rural Disparities

As shown in Table 4, rural children had more dietary energy intake than that of urban children in 1991 (102.2 kcal/day), while urban children had more in 2004 (103.9 kcal/day), and this difference increased further in 2015 (113.2 kcal/day). The urban-rural disparities in dietary fat and carbohydrate intake decreased over time across all age groups and gender (*p* < 0.001). Generally, rural children tend to have more dietary fat intake than their urban counterparts, while the distribution of carbohydrate intake was reversed. In 2015, the average urban-rural disparities in dietary fat and carbohydrate intake were 7.0 g/day and −5.3 g/day in Chinese children, respectively. The largest urban-rural disparities of fat (8.3 g/day) and carbohydrate intake (−6.7 g/day) was observed in boys aged 12 to 17 years. It is worth noting that urban children consumed more protein than that of rural children, and the difference increased significantly over time (*p* < 0.001). In 2015, the average urban–rural disparity in dietary protein intake was 7.6 g/day in Chinese children.

## 4. Discussion

Using more than two decades of data from the CHNS, the present study sought to determine the nutritional status of Chinese children, which is concentrated in urban–rural differences, through a longitudinal study related to nutritional status. China has made considerable progress in the nutrition of children, and the trend of energy intake has been declining in each round of investigation. However, Chinese children experienced a fast nutritional transition to a low-carbohydrate and high-fat diet. Meanwhile, obvious urban–rural disparities persistently existed in the distribution of dietary macronutrients composition.

The analysis of CHNS data found an overall decrease in energy intakes among children at 6 to 17 years of age from 1991 to 2015, which is consistent with similar studies in other age groups (adults and elderly) in China [10,11]. Moreover, the current results are consistent with a previous study of children in the United States of America, which showed an estimated initial decline of 159 to 240 kcal/d at the median observed across all age groups [24], while another survey showed that the daily mean dietary energy intake exceeded the dietary recommendations across all gender-age groups of African American children in Baltimore city [25]. In our research, we found that the decrease in the energy intake may be due to the decrease in the energy consumption, as sedentary behaviors are common in the daily lives of modern Chinese children [26]. Findings from the Youth Study in China (2016) showed that approximately 70% of school-aged children did not meet the moderate-to-vigorous physical activity recommendations (60 min/day) [27]. There was also evidence that along with the dramatic changes occurring in China, there has been an increase in the physical inactivity of children in both urban and rural areas [28]. The present study also found that energy intake was higher in urban children in 1991, whereas the opposite was true in 2004 and thereafter. It is widely acknowledged that educational facilities are essential public service facilities in rural China [29]. Due to inadequate basic educational facilities in 1991, rural children living farther away from school needed to walk/cycle longer distances daily, which may have resulted in greater energy expenditure, requiring a higher energy intake [30].

The scientific evidence is clear that a high-fat diet relates to chronic health problems such as heart disease, cancer, diabetes, and obesity [31]. Findings from the current analysis reported an overall increase in fat intakes and the % of energy from fat among children. In 2015, the mean average % of energy from fat in Chinese children was 36.8%, which exceeded the proportion of 30% recommended in Chinese dietary guidelines. This result is in line with a previous cross-sectional study conducted in China, which reported that the % of energy from fat was 36.8% in children aged 4 to 17 years [32]. Moreover, we found a significant increase over time in the proportion of children who consumed more than 30% of their energy from fat and less than 50% of energy from carbohydrates. The findings are consistent with results from Chinese young children (aged two to six years), which showed nutritional transition to a low-carbohydrate and high-fat diet during 2000 to 2011 [12]. Similarly, findings form the Korea National Health and Examination Survey (2007 to 2017) reported that fat consumption has increased, whereas the carbohydrate intake has declined among Korean adolescents [33]. However, a previous study in German children reported that fat quantity and quality did not change substantially between 2000 and 2010 [34]. In addition, we found that urban children usually consume more fat than rural children, similar to other findings in China [31], Cambodia [35] and Thailand [36]. With the high pace of economic development, parents nowadays are more likely to buy high-fat food, which may lead to children’s rapid weight gain. Protein is considered a key nutritional component in a well-balanced diet because protein influences growth in early life. The average dietary protein intake of children in this study (51.1 g/day) was close to that of school-age children in seven cities of China (50.2 g/day). Moreover, we found a decreasing trend in dietary protein intake among Chinese children over time. The causes of the decrease in protein intake are difficult to determine and can only be conjectured. As the proportion of the protein intake contributed by plant food is 47.8% [37] and the top two foods among plant food are pulses and rice, a possible explanation was that the amount of food from plants, such as rice and beans, decreased significantly. More studies are needed to explore other possible causes. Moreover, it should be noted that urban-rural disparities in the dietary protein intake increased notably, regardless of age and gender. Considering the relatively higher education level of parents in the current study, urban children probably have more high-quality protein (animal foods, milk and eggs) intake. Therefore, the Chinese government needed to take effective measures to promote the healthy diet of Chinese children, taking into account the accelerated change of nutrition and epidemiology and the increase of the burden of NCD_S_.

China has made rapid economic growth and great urbanization progress over the past 24 years, and its residents are also undergoing a dramatic nutritional transformation. In 1992, the International Conference on Nutrition adopted the World Declaration and Plan of Action for Nutrition, which is a vital step in the direction of a truly global commitment to action by countries and the international community alike [38]. Since then, the Chinese government promulgated a series of policies to improve the nutritional status of children, including the Action Plan for the Improvement of Children’s Nutrition in China (1996 to 2000), the Program for the Development of Food and Nutrition in China (2001 to 2010), and the National Children’s Development Plan (2011 to 2020) [39,40]. Furthermore, the Chinese government has made tremendous contributions to poverty reduction, increasing the construction of transportation systems and increasing agricultural production, which have the potential to increase the equity of access to all kinds of food for people of different backgrounds. Especially in the latest round of surveys, the percentages of energy from fat and carbohydrates were becoming closer between urban and rural areas. To better address the challenges posed by the nutrition transition, we recommend that governments further develop and refine targets and policies to promote healthy diets for children in urban and rural areas. Meanwhile, relevant services and guidance should be provided to create an environment that promotes children’s nutritional status.

Although this study is novel in some respects, some limitations should be considered when interpreting the results. First, dietary data in the present study was obtained by three consecutive 24-hour dietary recalls. While it can be used to evaluate the average amount of a population’s diet, random errors caused by variations in individual diets could lead to over- or underestimation. Nevertheless, the average intake in three days can offer a relatively valid estimate of nutrients intakes, as shown in an earlier study using the CHNS. Second, while CHNS does describe the health and nutritional status in China, it is not a nationally representative study. Third, macronutrients are found in a wide variety of foods. This study did not analyze the diversity of food groups and the specific type of each food group consumption, which made us unable to consider which kind of food led to the above deficiency.

## 5. Conclusions

Chinese children have experienced a fast transformation of nutrition to a diet with high fat and low carbohydrate. Public health policymakers should pay more attention to the problem of the excessive intake of total fat and high proportion of energy from total fat. Moreover, the urban–rural disparities in energy and macronutrient composition have persistently existed. To ensure that Chinese children have a more balanced diet, further nutrition education and intervention measures still required.

## Figures and Tables

**Figure 1 nutrients-13-01933-f001:**
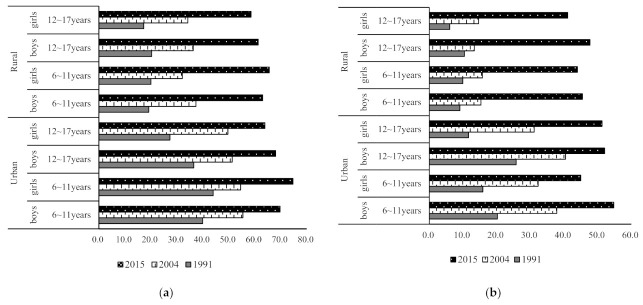
Trends of the proportion of children aged 6 to 17 years with more than 30% of energy from fat and less than 50% of energy from carbohydrates. (**a**) The proportion of children with more than 30% of energy from fat; (**b**) the proportion of children with less than 50% of energy from carbohydrates.

**Figure 2 nutrients-13-01933-f002:**
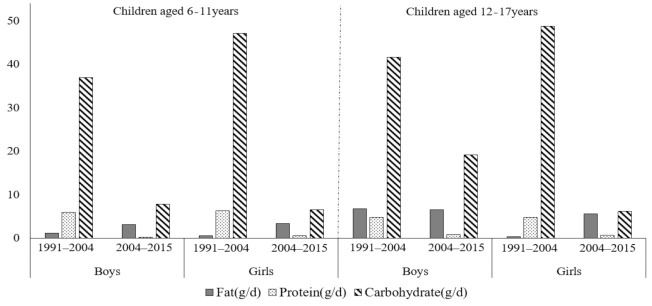
Comparison of the change values of urban-rural differences in dietary fat, protein and carbohydrate intake in each of the two rounds of surveys.

**Table 1 nutrients-13-01933-t001:** Sociodemographic characteristics of participants by CHNS rounds, 1991 to 2015 ^1–3^.

	1991	2004	2015	*p*-Value
Child Characteristic				
Sample (*N*)	2887	1979	2699	-
Age group (%)				
6 to 11 years	51.2 (0.9)	56.0 (1.1)	57.4 (0.9)	<0.001
12 to 17 years	48.8 (0.9)	44.0 (1.1)	42.7 (0.9)	<0.001
Gender (%)				
boys	50.7 (0.9)	48.3 (0.9)	47.7 (1.1)	<0.001
girls	49.3 (0.9)	51.7 (1.1)	52.3 (1.0)	<0.001
Geographical region (%)	26.3 (0.1)	26.9 (0.1)	34.5 (0.1)	<0.001
urban	26.3 (0.1)	26.9 (0.1)	34.5 (0.1)	<0.001
rural	73.7 (0.1)	73.1 (0.1)	65.5 (0.1)	<0.001
BMI (kg/m^2^)	17.1 (0.1)	17.9 (0.1)	18.5 (0.2)	<0.001
PA (MET/s)	85.0 (1.6)	84.6 (1.9)	70.2 (1.8)	<0.001
Household characteristic ^4^				
Education (%)				
primary/below	58.8 (0.9)	20.6 (0.9)	11.6 (0.6)	0.000
middle	26.4 (0.8)	54.7 (1.1)	42.0 (0.9)	0.000
high/above	14.8 (0.7)	24.7 (0.1)	46.4 (0.1)	0.000
Income	4.8 (0.1)	10.6 (0.3)	43.7 (4.1)	0.000
Smoking	65.6 (0. 9)	60.0 (1.1)	45.9 (1.2)	0.000
Drinking	63.9 (0.9)	40.0 (1.1)	46.1 (1.2)	0.000

^1^ Mean percentage ± standard error (SE). ^2^ CHNS, China health and nutrition survey.^3^ Significant trend in each subgroup across the survey years.^4^ Education, the highest educational level of the parents. Income, average annual household income. Smoking, parents have smoking behaviors. Drinking, parents have drinking behaviors.

**Table 2 nutrients-13-01933-t002:** Trends of the daily energy and macronutrient composition of children aged 6 to 17 years by CHNS rounds, 1991 to 2015 ^1–4^.

	1991	2004	2015	*p*-Value
Energy (kcal/day)	2205.3 (12.3)	1855.6 (15.5)	1671.6 (15.1)	<0.001
Fat (g/day)	55.6 (0.6)	58.5 (0.9)	66.2 (0.8)	<0.001
Protein (g/day)	63.5 (0.4)	57.2 (0.6)	51.1 (0.5)	<0.001
Carbohydrate (g/day)	369.8 (2.3)	274.7 (2.4)	247.6 (2.5)	<0.001
Fat (En%)	24.2 (0.2)	27.4 (0.3)	36.8 (0.3)	<0.001
Carbohydrate (En%)	63.4 (0.2)	60.1 (0.3)	51.4 (0.3)	<0.001

^1^ Mean percentage ± standard error (SE). ^2^ CHNS, China health and nutrition survey. En%, % of energy. ^3^ Values adjusted for age groups, gender and geographical regions, BMI and PA of children and household characteristics. ^4^ Significant trends in each sub-group across all survey years.

**Table 3 nutrients-13-01933-t003:** Trends in the daily energy and macronutrient composition by region, gender and age group from CHNS round, 1991–2015 ^1–4^.

	Urban								Rural							
	Boys				Girls				Boys				Girls			
	1991	2004	2015	*p*-Trend	1991	2004	2015	*p*-Trend	1991	2004	2015	*p*-Trend	1991	2004	2015	*p*-Trend
6 to 11 years																
Energy(k kcal/day)	1.9(0.1)	1.8(0.1)	1.6(0.1)	<0.001	1.8(0.1)	1.6(0.1)	1.5(0.1)	<0.001	2.0(0.1)	1.7(0.1)	1.5(0.1)	<0.001	1.9(0.1)	1.5(0.1)	1.4(0.1)	<0.001
Fat(g/day)	56.0(2.2)	62.2(3.8)	63.9(2.5)	<0.001	56.8(2.6)	58.6(3.3)	66.0(2.9)	<0.001	46.1(1.3)	51.4(1.7)	59.2(1.7)	<0.001	43.7(1.3)	44.0(1.7)	57.9(1.6)	<0.001
Protein(g/day)	58.7(1.3)	58.3(2.8)	51.7(1.4)	<0.001	53.6(1.2)	52.1(1.8)	50.9(2.0)	<0.001	57.8(0.8)	51.4(1.1)	44.6(0.9)	<0.001	55.5(0.8)	44.5(1.1)	43.4(1.0)	<0.001
Carbohydrate(g/day)	293.1(5.1)	244.1(2.2)	197.8(2.2)	<0.001	268.9(6.2)	218.8(7.5)	190.0(8.6)	<0.001	343.7(4.3)	257.8(5.2)	191.9(4.9)	<0.001	327.3(4.5)	230.1(4.7)	185.6(4.9)	<0.001
Fat(En%)	27.2(0.8)	33.2(1.5)	36.7(0.7)	<0.001	29.2(0.8)	33.3(1.3)	37.8(0.8)	<0.001	21.5(0.5)	26.2(0.6)	35.7(0.6)	<0.001	21.6(1.5)	25.1(0.7)	36.0(0.6)	<0.001
Carbohydrate(En%)	59.9(0.8)	53.8(1.5)	49.9(0.8)	<0.001	58.1(0.8)	53.5(1.3)	48.9(0.8)	<0.001	66.3(0.5)	61.5(0.6)	51.9(0.6)	<0.001	65.9(0.5)	62.8(0.7)	51.6(0.6)	<0.001
12 to 17 years																
Energy(k kcal/day)	2.6(0.1)	2.2(0.1)	2.0(0.1)	<0.001	2.2(0.1)	1.9(0.1)	1.7(0.1)	<0.001	2.6(0.1)	2.2(0.1)	1.9(0.1)	<0.001	2.4(0.1)	1.8(0.1)	1.7(0.1)	<0.001
Fat(g/day)	76.1(3.8)	78.7(3.6)	80.1(3.5)	<0.001	62.4(3.1)	68.4(3.2)	66.4(3.3)	<0.001	54.6(1.4)	63.9(2.0)	71.8(2.6)	<0.001	50.4(1.4)	56.1(1.8)	64.6(1.3)	<0.001
Protein(g/day)	75.8(1.5)	73.9(2.6)	65.8(1.9)	<0.001	64.8(1.5)	65.8(3.2)	57.5(2.5)	<0.001	73.9(1.0)	65.3(1.3)	56.4(1.5)	<0.001	67.9(0.9)	54.1(1.0)	49.3(1.3)	<0.001
Carbohydrate(g/day)	395.3(7.1)	310.8(8.1)	252.0(8.0)	<0.001	337.3(6.5)	265.1(7.0)	215.4(1.5)	<0.001	462.9(6.2)	366.7(5.5)	245.2(7.3)	<0.001	417.1(5.2)	276.1(4.3)	226.2(7.5)	<0.001
Fat(En%)	26.4(0.8)	30.7(0.9)	35.6(0.9)	<0.001	25.1(0.8)	29.8(1.0)	35.1(1.1)	<0.001	22.1(0.5)	26.0(0.7)	34.7(0.8)	<0.001	20.7(0.5)	25.9(0.7)	34.1(0.9)	<0.001
Carbohydrate(En%)	61.1(0.8)	56.2(0.9)	51.3(0.9)	<0.001	62.2(0.8)	56.8(0.9)	51.0(1.0)	<0.001	65.9(0.5)	61.9(0.7)	52.8(0.8)	<0.001	67.2(0.5)	62.0(0.7)	53.6(0.8)	<0.001

^1^ Mean percentage ± standard error (SE). ^2^ CHNS, China health and nutrition survey. En%, % of energy. ^3^ Values adjusted for BMI and PA of children and household characteristics.^4^ Significant trends in each sub-group across all survey years. A significant increase in the proportion of children with more than 30% of energy from fat and less than 50% of energy from carbohydrates were observed in this study (*p* < 0.001). In 2015, the proportion of children aged 6 to 17 years with higher fat intake and lower carbohydrate intakes compared with the DRI were 64.7% and 46.8%, respectively. Specifically, from 1991 to 2015, the proportion with higher fat intake increased from 34.3 % to 69.8 % in urban children, and increased from 18.9 % to 62.8% in rural children, respectively. Meanwhile, the proportion with lower carbohydrate intake increased from 18.6 % to 51.0 % in urban children, and from 8.6 % to 45.2% in rural children, respectively. As seen from Figure 1, urban children aged 6 to 11 years had the highest proportion who consumed more than 30% of energy from fat (69.7% in boys and 74.7% in girls) in 2015.

**Table 4 nutrients-13-01933-t004:** Trends of the urban–rural disparities in the daily energy and macronutrient composition by gender and age group from CHNS round, 1991~2015 ^1–5^.

	Boys				Girls				Overall			
	1991	2004	2015	*p*-Trend	1991	2004	2015	*p*-Trend	1991	2004	2015	*p*-Trend
6 to 11 years												
Energy(kcal/day)	−109.0(4.5)	69.6(7.4)	91.1(5.6)	<0.001	−122.5(4.4)	116.9(5.3)	123.1(1.6)	0.001	−102.2(3.5)	103.9(4.4)	113.2(3.5)	<0.001
Fat(g/day)	9.9(2.5)	8.8(0.8)	5.7(0.3)	<0.001	11.1(2.3)	10.6(3.0)	7.2(0.7)	0.001	11.8(0.7)	10.3(0.4)	6.6(1.9)	<0.001
Protein(g/day)	1.0(0.1)	6.9(2.6)	7.1(0.7)	<0.001	0.9(1.4)	7.2(1.8)	7.7(0.7)	0.001	0.9(0.1)	7.1(1.5)	7.5(1.2)	<0.001
Carbohydrate(g/day)	−50.6(1.1)	−13.7(1.1)	−5.9(0.7)	<0.001	−58.3(1.8)	−11.2(1.0)	−4.7(0.9)	0.001	−52.2(2.6)	−11.4(1.1)	−5.2(0.9)	<0.001
Fat(En%)	5.7(0.9)	5.0(0.5)	0.9(0.1)	<0.001	6.6(0.9)	6.2(1.3)	1.8(0.9)	0.001	6.6(0.6)	5.7(1.0)	1.3(0.1)	<0.001
Carbohydrate(En%)	−6.5(0.9)	−5.8(1.5)	−2.0(0.9)	<0.001	−7.8(0.9)	−7.3(1.3)	−2.7(0.9)	0.001	−7.1(0.6)	−6.7(1.0)	−2.3(0.1)	<0.001
12 to 17 years												
Energy(kcal/day)	−70.4(6.1)	63.8(4.4)	126.5(1.0)	<0.001	−123.4(3.1)	73.2(4.7)	116.6(0.4)	0.001	−92.0(3.9)	68.8(4.1)	122.3(5.5)	<0.001
Fat(g/day)	21.5(3.2)	14.8(2.8)	8.3(2.2)	<0.001	12.0(2.7)	12.3(3.1)	6.7(0.1)	0.001	17.8(2.1)	13.2(2.4)	7.4(1.1)	<0.001
Protein(g/day)	1.9(0.1)	6.6(0.6)	7.4(2.2)	<0.001	3.0(0.7)	7.7(2.1)	8.3(2.2)	0.001	2.8(0.3)	6.8(0.7)	7.7(0.6)	<0.001
Carbohydrate(g/day)	−67.5(1.1)	−25.9(0.7)	−6.7(1.0)	<0.001	−59.8(1.7)	−11.0(1.2)	−4.8(1.2)	0.001	−63.9(0.2)	−7.3(0.6)	−5.4(0.9)	<0.001
Fat(En%)	4.9(0.1)	4.4(0.2)	0.9(0.2)	<0.001	4.4(0.9)	3.8(0.1)	1.1(0.1)	0.001	4.6(0.6)	4.3(1.8)	1.1(0.9)	<0.001
Carbohydrate(En%)	−5.1(0.9)	−4.7(1.1)	−2.1(0.2)	<0.001	−5.3(0.9)	−5.2(1.1)	−2.6(0.3)	0.001	−5.2(0.6)	−5.0(0.8)	−2.2(0.9)	<0.001

^1^ Mean percentage ± standard error (SE). ^2^ CHNS, China health and nutrition survey. En%, % of energy. ^3^ Values adjusted for BMI and PA of children and household characteristics. ^4^ Significant trends in each sub-group across all survey years. ^5^ Negative number means urban is smaller than rural. Comparison of the change values of urban–rural differences in dietary fat, protein and carbohydrate intake in each two rounds of surveys were presented in Figure 2. Generally, the variation of dietary protein and carbohydrate intake between urban and rural areas was greater in the former two rounds (2004 vs. 1991) than in the latter two rounds (2015 vs. 2004), while the variation in fat intake among children aged 6 to 11 years was the opposite (*p* < 0.001). The average variation of dietary protein and carbohydrate intake decreased from 9.5 g/day and 32.8 g/day in the former two rounds to 1.1 g/day and 13.7 g/day in the latter two rounds, respectively. Meanwhile, the variation of urban-rural disparities in dietary fat intake increased from 1.1 g/day to 3.1 g/day among boys and from 0.5 g/day to 3.4 g/day among girls aged 6 to 11 years, respectively.

## Data Availability

The datasets generated and/or analyzed during the current study are not publicly available but are available from the corresponding author on reasonable request.

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
