# Peer review of "Trends and Urban-Rural Disparities of Energy Intake and Macronutrient Composition among Chinese Children: Findings from the China Health and Nutrition Survey (1991 to 2015)"

_nutrients, 2021, doi:10.3390/nu13061933_

Round 1
Reviewer 1 Report
See attached

Author Response
Please see in the attachment

Reviewer 2 Report
Thank you for the opportunity to review this manuscript, which investigates secular trends in dietary energy, macronutrient intake, and macronutrient distribution of Chinese children, as well as urban-rural disparities in children’s nutrition, from 1991-2015. Strengths include a large sample size, longitudinal data, and a robust study. Main weaknesses included a weak Introduction and some English language errors.
General
1) Please review the manuscript for English language errors. Overall, I understand what you’re communicating. However, there are multiple grammatical issues.
- Missing punctuation: As one example, there is a missing comma on line 35 (“…nutrients may lead to malnutrition, which affects…”).
- Incorrect word choice: As one example, line 39 would be clearer as “...children in China make up almost 14.8%…”
- Incomplete sentences/fragments: As one example, the sentence on lines 68-70 (“Using data … of its population [12].”) is not a complete sentence.
These are only a few examples.
2) Relatedly, please check for verb tense usage and consistency. For example, please use present tense when referring to tables in your manuscript (e.g., lines 112-113). Additional guidelines are available here: https://www.unl.edu/gradstudies/connections/writing-about-your-research-verb-tense.
3) You used “NCD” only three times in your manuscript (i.e., lines 38, 214, and 243). Instead of abbreviating it, please write out “non-communicable diseases.” Relatedly, you introduce “%En” on lines 135-138, but fail to use that abbreviation for “% of energy” elsewhere in your manuscript (e.g., line 22, line 142). Because “%En” is used only four times, I recommend writing out “% of energy” instead.
Introduction
4) Overall, the Introduction is underdeveloped and should better set up the study. What does the existing literature reveal about energy intake and macronutrient composition in Chinese children? What does the existing literature reveal about rural and urban differences? Why choose 6-17 year olds? The Introduction should reveal what is already known and then segue into the gaps (which you have on lines 54-59).
5) Lines 60-65: Please rewrite the paragraph on lines 60-65. As is, you have one long sentence, which should be broken up and checked for grammar/punctuation. In addition, “supply the possible explanation and potential dietary recommendations for nutritional indicators of children in China” is not a purpose of your study: you didn’t collect or analyze data related to that.
Results
6) In your tables, please use “<0.001” instead of “0.000” for the p-values.
7) Table 1: Please check the alignment. Specifically, the data for the urban and rural rows are off, and there are missing data for the BMI row.
Discussion
8) In the Discussion, you compare your findings to U.S. children (lines 195-199), Korean children (lines 2018-221), German children (lines 221-222) and Cambodian and Spanish children (lines 222-224). It is difficult to compare across countries, given differences in cultures, diets, development, etc. I recommend focusing on research/data among Chinese children.
9) Lines 191-192: You write “Further nutritional interventions … for Chinese children” as if this is a focus or finding of your study. It is not. This sentence, however, could be included and expanded as a separate paragraph in the Discussion to help explain your findings.
10) Lines 193-195: You could include more information about the obesity epidemic and “the rapid increase in the prevalence of overweight and obesity in children in China” in the Introduction. This could help set up your study.
11) Lines 244-250: Please expand on this paragraph, the contributions of the Chinese government, and other efforts related to nutritional interventions and education. This will help readers understand the context and what was happening in China from 1991-2015 to explain your study’s findings. Also, please add citations/references to support this discussion.
Author Response
Please see the attachement
